# Biocompatibility Analysis of GelMa Hydrogel and Silastic RTV 9161 Elastomer for Encapsulation of Electronic Devices for Subdermal Implantable Devices

David Catalin Dragomir [1,2], Vlad Carbunaru [3,*], Carmen Aura Moldovan [1], Ioan Lascar [3,4], Octavian Dontu [2], Violeta Ristoiu [5], Roxana Gheorghe [5], Ana Maria Oproiu [3,4], Bogdan Firtat [1], Eduard Franti [1,6], Monica Dascalu [2,6], Tiberiu Paul Neagu [3,4], Dan Mircea Enescu [4], Octavian Ionescu [1,7], Marian Ion [1], Carmen Mihailescu [1], Ruxandra Costea [8], Magda Gonciarov [8], Gabriela Ionescu [7], Adrian Dumitru [4], Anca Minca [4], Catalin Niculae [9], Stefania Raita [8], Ioana Rosca [4], Sorin Lazarescu [4], Cristian Stoica [4], Raluca Ioana Teleanu [4] and Daniel Mihai Teleanu [4]

1    IMT Bucharest, 77190 Bucharest, Romania
2    Faculty of Mechanical Engineering and Mechatronics, University Politehnica of Bucharest, Splaiul Independentei 313, 060042 Bucharest, Romania
3    Emergency University Hospital Bucharest, Calea Floreasca 8, 014461 Bucharest, Romania
4    Faculty of Medicine, University of Medicine and Pharmacy "Carol Davila" Bucharest, Bulevardul Eroii Sanitari 8, 050474 Bucharest, Romania
5    Faculty of Biology, University of Bucharest, Splaiul Independenţei 91-95, 050095 Bucharest, Romania
6    Center for New Electronic Architecture, Romanian Academy Center for Artificial Intelligence, 050711 Bucharest, Romania
7    Engineering Mechanical and Electrical Faculty, Petroleum and Gas University from Ploiesti, 100680 Ploiesti, Romania
8    Faculty of Veterinary Medicine, University of Agricultural Sciences and Veterinary Medicine of Bucharest, Splaiul Independentei, nr. 105, 050097 Bucharest, Romania
9    AREUS Technology SRL, Bd. Chisinau 10, 011687 Bucharest, Romania
*    Correspondence: vlad.carbunaru@drd.umfcd.ro

**Abstract:** The natural differences between human-made electronics and biological tissues constitute a huge challenge in materials and the manufacturing of next-generation bioelectronics. As such, we performed a series of consecutive experiments for testing the biofunctionality and biocompatibility for device implantation, by changing the exterior chemical and physical properties of electronics coating it with silicone or hydrogels. In this article, we present a comparison of the main characteristics of an electronic device coated with either silicone or hydrogel (GelMa). The coating was performed with a bioprinter for accurate silicone and hydrogel deposition around different electronic chips (Step-Down Voltage Regulator U3V15F5 from Pololu Corporation). The results demonstrate that the hydrogel coating presents an augmented biomechanical and biochemical interface and superior biocompatibility, lowers foreign body response, and considerably extends the capabilities for bioelectronic applications.

**Keywords:** implantable; electronic devices; coating; biocompatible; hydrogel

## 1. Introduction

The last decade heralded the appearance of high-performance, multipurpose, and printable electronic devices, which have propelled the development of a brand new age of biomedical devices: bioelectronics. These platforms combine the biologic and the electronic onto a unique stage, and consequently offer new capabilities for implantable devices and organ–electronics interfaces [1].

Despite the extraordinary advances within recent decades, the natural differences between human-made electronics and biological tissues constitute a huge challenge in materials, layout, and manufacturing of next-generation bioelectronics. Nearly all laboratory-

degree commercial bioelectronics are manufactured from dry and rigid components such as silicon and metals, in contrast to the human body, which contains a wide range of water-holding soft tissues and organs [2,3]. Such obvious dissimilarity between the two presents sizeable problems regarding the organ–electronics interface. We are addressing this issue by changing the exterior chemical and physical properties through encapsulation in hydrogels–water-infiltrated crosslinked polymers.

These hydrogels have recently captivated a growing interest in bioelectronics due to their similarity to biological tissues [4,5]. The exceptional flexibility of their biological, mechanical, and electrical properties offers hydrogels a unique bridging potential of the material–biological world. This is the new extensively studied and reviewed trend in tissue engineering, in an attempt to create a "flawless" interface [6].

We optimized the deposition of biologic and electronic materials in order to achieve a single platform with complete and full spatial control via extrusion-based printing of gelatin methacryloyl (GelMA) hydrogel [6] and medical silicone. Bioprinting provides accurate and rapid fabrication, with interfaces accessible immediately after printing, consequently requiring no post-processing.

In the following study, we address some of the concerns in both biocompatibility and functionality of free-form GelMA bioelectronic platforms through comparison with medical silicone, the most commonly used material for coating implantable devices, from different types of catheters and needles to breast prostheses and heart simulators. Silicone is suitable for this study because it cures at room temperature, can be used at temperatures ranging from $-50\ ^\circ$C to $250\ ^\circ$C, has excellent dielectric properties, and is highly resistant to moisture and oxidation, which is very good for protecting the electronic device from malfunctioning while in vivo.

We performed a series of consecutive experiments, starting from the biofunctional standpoint, by measuring the conductibility and resistance of both encapsulated devices, followed by analyzing their biocompatibility in a controlled in vitro experiment using RAW 264.7 macrophages. Ultimately, we implanted both silicon- and GelMA-coated devices into a living animal (*Sus scrofa*) for 21 days, and the selected tissue samples around the implanted devices were histopathologically analyzed in order to corroborated the results.

Our aim was to design platforms where biologic and electronic materials coexist in synergy and add to the existing knowledge by exploiting the fabrication process of 3D bioprinting.

## 2. Materials and Methods

### 2.1. Preparation of GelMA and Photoinitiation

GelMA, or methacrylated gelatin, was synthesized by adding methacrylic anhydride in a 10% ($w/v$) suspension of gelatin at a rate of 0.5 mL/min under steady stirring, which was previously mixed with phosphate buffer saline (PBS). Materials were supplied by Cell Ink, Gothenburg, Sweden. After 3 h at 50 $^\circ$C, the mixture was dialyzed in distilled water for 7 days at 40+ $^\circ$C in order to separate methacrylic acid and anhydride. Finally, the solution was sieved through a 0.22 μm membrane in order to obtain pure GelMA, which was stored at $-20\ ^\circ$C until usage [7,8].

The final step in the hydrogel's formula was the integration of lithium phenyl-2,4,6-trimethylbenzoylphosphinate (LAP) photoinitiator. LAP, at 0.5% concentration ($w/v$), is designed to crosslink photocurable hydrogels and bioinks using the photocuring module with UV light. LAP absorbs light in the ultraviolet spectral range, in this case, at a 405 nm specter, and converts this light into chemical energy in the form of reactive intermediates such as free radicals and reactive cations, which subsequently initiate polymerization.

### 2.2. Preparation of Medical Silicone

The proposed elastomer RTV 9161 silicone rubber is a suitable material that has medical grade applications. This is a bicomponent material that starts curing when mixed with Dow Corning catalyst 9162 in a minimum ratio of 20:1 (elastomer:catalyst). The concentrations of the catalyst depend on the application and final needed elasticity. It has

been widely used for the fabrication of sterilized medical-grade tubing used in hospitals. After being mixed with catalyst and applied to the desired device, it is recommended to thermally treat the material above the maximum operating temperature of the device for at least 4 h.

### 2.3. Electronic Device

In order to test the hydrogel-encapsulated device in terms of conductivity and to make sure that a short circuit was not created by the material touching electrical paths, conductivity measurements were performed on a small electronic device. The targeted electronic device was a 5 V, 600 mA Step-Down Voltage Regulator U3V15F5 from Pololu Corporation (Las Vegas, Nevada, USA). Because input contacts and certain SMD component pins are not isolated with specific electronics' green coating, measuring conductivity by testing electric resistance through the device is relevant. The device was approximately $0.32'' \times 0.515'' \times 0.1''$, weighing 0.4 g. It had a minimum operating voltage of 2.5 V and a maximum of 5 V, with a maximum input current of 1.4 A and output voltage of 5 V.

### 2.4. Bioprinter

This printer BioX from CellInk, Sweden$^{TM}$ needs to function in a sterilized environment. The materials are loaded into special syringes to which printheads of different sizes starting from 0.1 to 0.5 mm are fitted (Figure 1). After the selected material has been loaded into the syringe and the printing head has been mounted, it is inserted into the extruder and secured by a mechanical coupling with a screw. The extrusion process is carried out by applying a low and constant pneumatic pressure to the end of the piston (syringe). When this pressure is higher than the internal pressure in which the hydrogel is located, extrusion start, and the extruder moves in the Cartesian space with CNC-type controls.

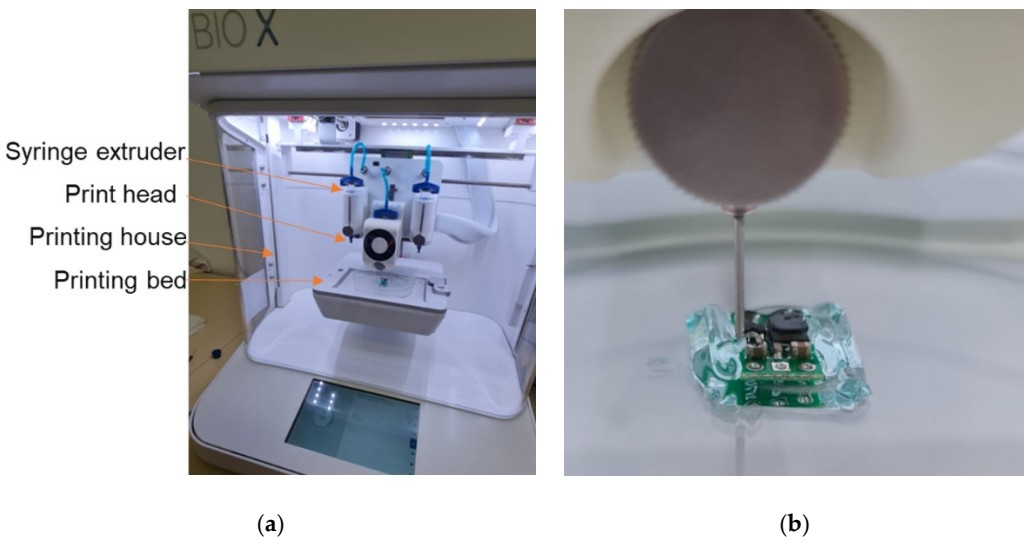

(**a**)　　　　　　　　　　　　　　　　　　　　　　　　　(**b**)

**Figure 1.** (**a**) Bioprinting process; (**b**) electronic device encapsulation with GelMa.

In order to meet the material's stability needs, the printer must provide very precise control of the temperature of the printing house, the printing bed, and the printhead. In this case, GelMa becomes liquid and reaches the viscosity of extrusion at pressures that can be reached by the printer at a fixed temperature of 26 +/− 0.2 °C. In order to strengthen and maintain its desired shape, the material must be cooled immediately after it has exited the print nozzle. Because the printed form is deposited on the printing bed, precise temperature control is necessary. The printing bed must have a temperature of 10 °C for the material to remain in its desired shape throughout the printing process. This material also incorporates particles of photoinitiators which activate in the UV light spectrum and

stabilize the material at the end of the manufacturing process so that shape is maintained even when the ambient temperature is reached.

### 2.5. GelMa Hydrogel Encapsulation

In order to cover the electric device with GelMA, we designed a prismatic shape, empty on the inside with a wall thickness of 1 mm, in dedicated CAD software (DNA Studio 4, Gothenburg, Sweden). When the printing process reached the stage where lid piece printing started, the process was paused, the electronic device was manually inserted, and the printing process resumed to achieve the sealed encapsulation.

We loaded the GelMa hydrogel into a syringe and fit a 0.53 mm print nozzle (27 Gauge). A specific set of values must be meticulously followed in order to obtain stable results: extrusion pressure of 65 kPa; temperature of the printhead of 26 °C; temperature of the printing bed of 10 °C; printing speed of 20 mm/s; and the form fill factor of 90%. UV light treatment at 405 nm was applied to every 3 layers deposited at a distance of 5 mm from the workpiece for 10 s, followed by a treatment of 10 min with the same parameters at the end of the process (Figure 2).

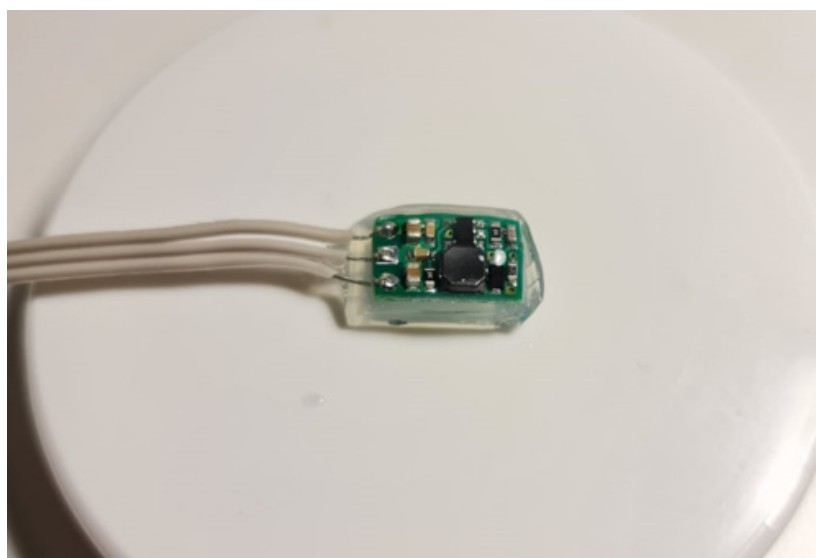

**Figure 2.** GelMa-encapsulated device.

After the electronic device was encapsulated, it was soaked in the crosslinking solution provided by Celink (which contained 50 mM calcium chloride to crosslink bioinks containing alginate) [9] to further increase its electronic and mechanical stability.

### 2.6. Silastic RTV 9161 Medical-Grade Silicone Encapsulation

The same device type was encapsulated with medical-grade silicone Silastic RTV 9161, with no conductivity issue due to its zero water content. The conductivity tests performed before and after encapsulation showed minor modification in resistance value (Figure 3).

### 2.7. In Vitro Experiment

The materials were washed with sterile phosphate-buffered saline (PBS) and then left overnight at 4 °C, in 10% penicillin/streptomycin (P4458; Sigma, Saint Louis, MS, USA) solution and, the next day, were sterilized by exposure to UV light. RAW 264.7 macrophages (kindly donated by Prof. Anisoara Cimpean) were seeded at $2 \times 10^4$ cells/mL, cultured in standard conditions (37 °C, 5% $CO_2$ and humidity), and then the biocompatibility assay was performed at 3 and 6 days post-seeding.

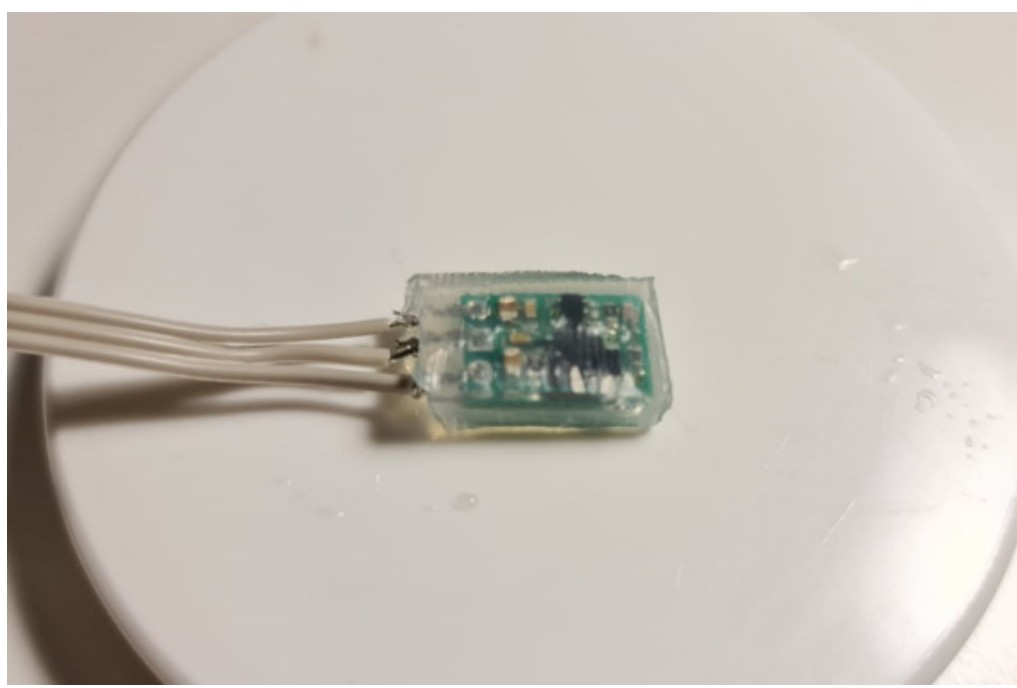

**Figure 3.** RTV 9161 elastomer-encapsulated device.

For the biocompatibility evaluation, GelMA- or silicone-covered circuits and the conductive wire used for the circuits were compared with the tissue culture plate as a control.

**Cell culture.** The RAW 264.7 cell line (macrophage-like cells) were cultured in Dulbecco's modified Eagle's medium (31966-021; Gibco, Grand Island, New York, NY, USA) supplemented with 10% fetal bovine serum (10270106; Life Technologies, Waltham, MA, USA) and 1% penicillin/streptomycin (P4458; Sigma). Cells were passaged after reaching 90% confluence, detached with a cell scraper, and subcultivated for the experimental conditions. Cultures were maintained at 37 °C in a humidified 5% $CO_2$ incubator.

**Assay for cell viability.** RAW 264.7 macrophages were plated into 12-well plates ($2 \times 10^4$ cells/mL) in contact with the GelMA- or silicone-covered circuit and left for 3 and 6 days in culture. The RAW 264.7 cells' viability when in contact with the materials was determined by a 3-(4,5-dimethylthiazol- 2-yl)-2,5-diphenyltetrazolium bromide (MTT) spectrophotometric assay, which is based on the cleavage of the yellow tetrazolium salt to purple formazan crystals by mitochondrial and cytosolic dehydrogenases of living cells. Briefly, cells were incubated with MTT solution (1 mg/mL, M2128; Sigma) in serum and antibiotic-free culture medium for 3–4 h (at dark, standard conditions). The medium was then removed, and the formazan was solubilized by DMSO. Absorbance at 570 nm was measured with a microplate reader (FlexStation 3; Molecular Devices, Herrsching, Germany). Each experimental group was analyzed in triplicate, and the values are expressed as percentage of cell survival relative to control cells.

**Statistical Analysis.** All data are given as means ± SEM; statistical significance was tested using a 1-way ANOVA with a Bonferroni post-test (GraphPad Prism software (5.0, San Diego, California, SUA)). A value of $p < 0.05$ was considered to be statistically significant, with * $p < 0.05$, ** $p < 0.01$, and *** $p < 0.001$.

*2.8. In Vivo Experiment*

Both encapsulated devices were subdermal, behind the earlobe, implanted in a *Sus scrofa* (male, one-year-old) subject to the experiment. All procedures were approved by the Faculty of Veterinary Medicine Ethics Committee (Protocol no. 24 from 17 May 2022), and by the Romanian Sanitary Veterinary Directorate (Protocol no. 14 from 9 June 2022) and adhered to the guidelines set by the Public Health Service Policy on Humane Care and Use of Laboratory Animals (2015). After 3 weeks, both coated devices were

surgically extracted with the surrounding biological tissue for sectioning and transported to the pathology laboratory. The euthanasia protocol was carried out using T-61 solution under heavy anesthesia. The samples weighing approximately 20 g, retaining both their original implantation shape and size, and together with 2 cm of surrounding tissue, were immersed for 24 h with 10% buffered formalin. The samples were then prepared by conventional methods using paraffin embedding, sectioning and hematoxylin–eosin (HE) staining. Using biological samples formerly immersed in neutral formalin solution, the microscopic study was performed at ambient temperature, followed by paraffin embedding by the histopathological protocol. Additionally, the biological materials were incised and sectioned using a HM350 microtome outfitted with a hydrodisection transfer system (STS, microM). For the histological study, we trimmed the sections to a width of 2 μm and stained with hematoxylin–eosin (HE) [10]. All sections were analyzed using conventional brightfield microscopy (Panthera L Motic). Moreover, polarized light was used for a finer resolution of the foreign materials.

## 3. Results

### 3.1. Biofunctional Experiment

Conductivity measurements were performed using a UNI-T UT139A high-precision multimeter, by connecting the Vin and ground of the device with corresponding probes. The unencapsulated standard device had a resistance value of 612 Ohm, while the silicone-encapsulated device had a resistance value of 632 Ohm, and the GelMA-encapsulated device was measured at 697 Ohm.

Although voltage in (Vin) and the device's ground pins showed a slight difference compared with those of the unencapsulated device, in low-current devices, functionality is not affected.

### 3.2. Biocompatibility In Vitro Results

The results showed that the conductive wire, which is a component of the circuits, was nontoxic to the RAW 264.7 macrophages when compared with the control untreated cells, both after 3 days (Day 3: control untreated cells = 100 ± 0.1, *n* = 3; conductive wire = 100 ± 0.1, *n* = 3) and after 6 days in contact (Day 6: control untreated cells = 94 ± 4.6, *n* = 3; conductive wire = 102 ± 1.4, *n* = 3) (Figure 4).

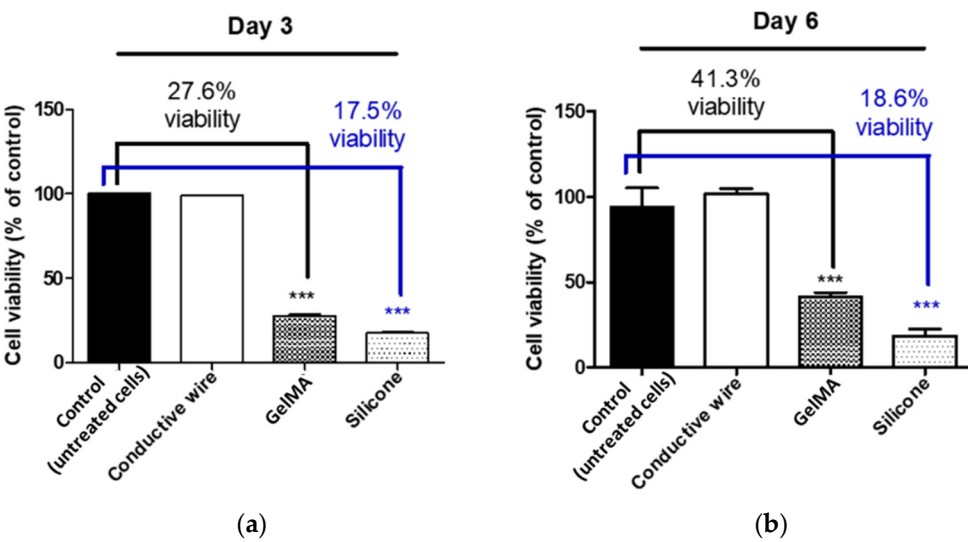

(**a**)                                                                (**b**)

**Figure 4.** Cell viability as revealed by MTT assay. RAW 264.7 cells biocompatibility profile after 3 days (**a**) or 6 days (**b**) in culture; a value of *p* < 0.05 was considered to be statistically significant, with * *p* < 0.05, ** *p* < 0.01, and *** *p* < 0.001.

As for the tested circuits, after 3 days in contact with the GelMA-treated circuit, the RAW 264.7 cells showed a viability of 27.6% compared with control untreated cells (Day 3: control untreated cells = $100 \pm 0.1$, $n = 3$; GelMA = $27.6 \pm 0.5$, $n = 3$), and this viability increased to 41.3% after 6 days in contact (Day 6: control untreated cells = $94 \pm 4.6$, $n = 3$; GelMA = $41.3 \pm 1.07$, $n = 3$) (Figure 4).

On the other hand, after 3 days in contact with the silicone-treated circuit, the RAW 264.7 cells showed a viability of only 17.5% compared with control untreated cells (Day 3: control untreated cells = $100 \pm 0.1$, $n = 3$; GelMA = $17.5 \pm 0.3$, $n = 3$), and this viability increased after 6 days in contact, but only to 18.6% (Day 6: control untreated cells = $94 \pm 4.6$, $n = 3$; GelMA = $18.6 \pm 1.8$, $n = 3$) (Figure 4).

*3.3. Biocompatibility In Vivo Results*

The coating of each device was examined using conventional brightfield microscopy (Panthera L Motic, Park Systems, Suwon, Korea) and polarized light, at ×20 and ×40 magnification, and HE staining. GelMA sections revealed a mixed inflammatory reaction with frequent macrophages, histiocytes (black arrows), and lymphocytes (yellow arrows) around fragments of partially integrated GelMA material (marked with green arrow in Figure 5a,b).

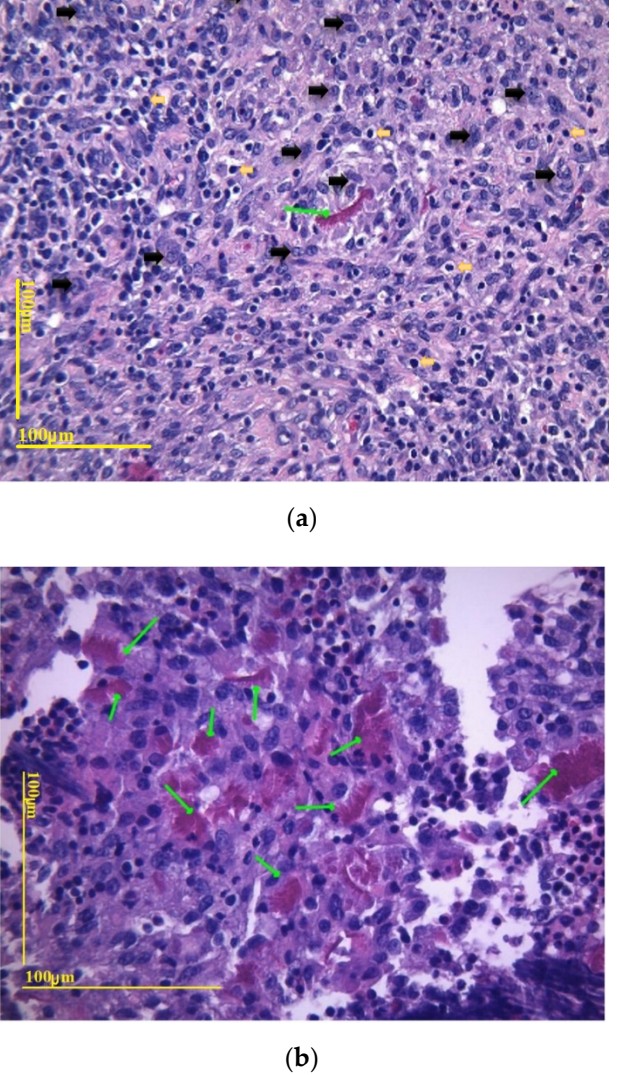

(**a**)

(**b**)

**Figure 5.** (**a**) GelMA section, ×20 magnification at 100 microns. Green arrows mark GelMA-biomaterial surrounded by histiocytes (black arrows) and lymphocytes (yellow arrows) random inflammatory cells; (**b**) GelMA section, ×40 magnification at 100 microns. Green arrows mark GelMA.

Meanwhile, the microscopic detail of the foreign body revealed an intense granulomatous reaction around the silicone fragments, with abundant eosinophils in the close vicinity of the foreign material (Figure 6a, red arrow for silicone, green arrows for macrophages, and purple arrows for eosinophils). Moreover, a different section revealed relatively frequent multinucleated foreign body giant macrophages trying to phagocytize (dismantle) the material (Figure 6b, red arrows—silicone; green arrows—macrophages). These aspects indicate the poor tissue integration of the foreign material, subpar in comparison with the GelMA integration.

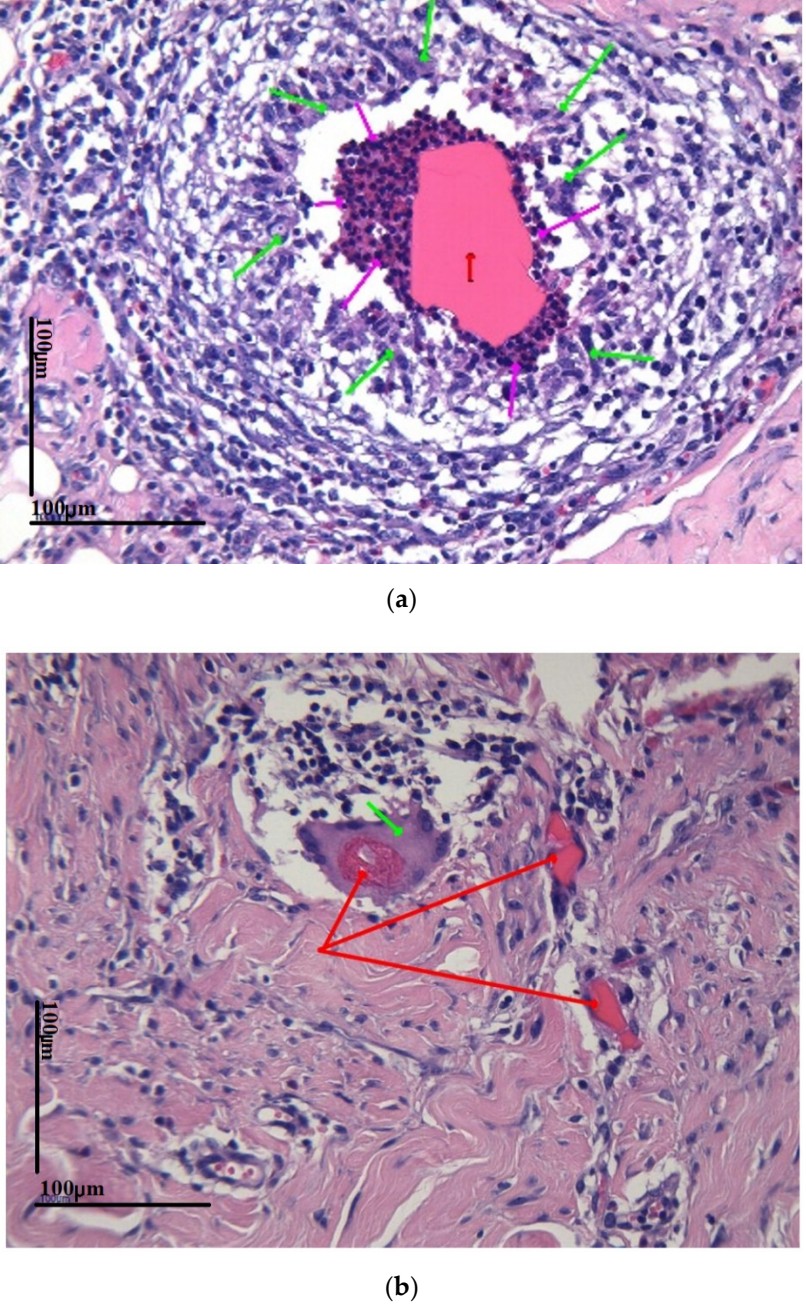

(**a**)

(**b**)

**Figure 6.** (**a**) HE staining, 20× magnification at 100 microns. Red arrows marks silicone. Green arrows mark macrophages. Purple arrows mark eosinophils; (**b**) HE stain 20× magnification at 100 microns. Green arrow marks a multinucleated foreign body giant cell trying to phagocytize the silicone (red arrow).

## 4. Discussion

Bioengineered platforms will slowly transform various antiquated methods of disease monitoring and be used to predict the efficacy of experimental therapeutics or interventions, possibly eliminating the need for animal subjects [11]. Nevertheless, conventional fabrication methods still produce simplistic builds or cell conditions. The emergence of 3D bioprinting enables the engineering of compound, biomimetic, in vitro models that aid the furtherance of disease therapy, along with management and care for patients.

Traditionally, it has been an unwritten understanding that silicone is intrinsically a biocompatible material because it has its myriad of purposes in healthcare applications. Its low surface tension and exceptional chemical and thermal stability have minimal adverse effects on its host. Despite its numerous advantages, it cannot support the cell viability function of adhesion-dependent cells as a consequence of being strongly hydrophobic and chemically inert [12]. As such, it will be consistently treated as a foreign body due to its cytotoxicity, with two variable outcomes: smaller pieces will be phagocytized by giant multinucleated foreign body cells, and larger pieces will be encapsulated by eosinophils, resulting in scarring and the formation of a capsule around the material. While mixed inflammatory reaction, with numerous lymphocytes, macrophages, and histiocytes can be observed in both materials, the presence of eosinophils around the silicone indicates inadequate tissue integration. In prolonged periods of embedment, cumulative biomechanical interactions and substantial incongruity of mechanical properties precipitate an excessive foreign body response with additional scarring [11,13]. Development of a thick scar that encapsulates the implantable electronic can deteriorate its performance by amplifying the interface's impedance and decreasing its stimulation or recording efficiency [14].

Further testing may be necessary, but following the biocompatibility experiment results, we concluded that the GelMA-coated circuit supports cell viability and cell integration within its structure. Similar work by Noshadi et al. showed that GelMA hydrogels supported the growth and functionality of primary cardiomyocytes in vitro and could promote tissue healing following a myocardial infarction in a rodent animal model in vivo [15]. GelMA-stable constructs were also successfully implanted in vivo in arteriovenous loops in syngeneic rats for 4 weeks, were well tolerated by the animal, and retained shape and size, showing that the slow degradation of GelMA is suitable for long-term implantation [16]. GelMa's superior biocompatibility bridges the mechanical and biological gap, by simulating the properties of the extracellular matrix. By mimicking the chemical properties of the extracellular matrix using GelMA encapsulation, a superior integration of bioelectronics is achievable [17–19]. As such, cell adhesion and proliferation along with tissue in-growth are nurtured on these scaffolds [20,21]. Owing to GelMA hydrogel's soft and malleable nature, mechanical-to-biological mismatch is minimized, while simultaneously providing a wet and ion-loaded environment, bestowing long-term stability in bioelectronics impedance.

Because hydrogels retain water as the main element in their composition, rigorous measurements are necessary to guarantee that the system works properly. It was closely observed that while curing the deposed GelMA, a strong polymerization happens at the exterior of hydrogel's surface and on the contact surface with the electronics. This happens because curing is performed in the UV spectrum, and, because the biomaterial is mostly transparent, the light will bounce off its borders and the surface that comes in contact with the encapsulated device, having its first effect there. This can be addressed as "deep curing" because UV light travels inside the hydrogel, but a second "surface curing" happens while submerging the encapsulated device in ionic crosslinking solution [22–25]. Such diverse composition and chemical treatment may be the reason for higher electric resistance in GelMA encapsulation than in the silicone-encapsulated device.

The silicone is a bicomponent material, and, because the curing agent is thoroughly mixed in all its composition, curing is uniformly activated across the whole volume. The lower resistance value shows that the internal chemical composition of the silicone has elements that favor a higher electric flow than GelMA encapsulation.

The resistance values of the hydrogel-encapsulated device are higher than the one encapsulated with silicone, but the difference is not big enough to lead to a malfunctioning device. After curing GelMa hydrogels with UV light and crosslinking solutions, applying it to electronic devices does not interfere with low-current devices.

Despite all this progress, most of the work on hydrogel bioelectronics has been concentrated on material developments, followed by proof-of-concept-type experimentation. As this field evolves beyond its early stage, it offers plenty of space for future developments. Further improvement in hydrogels' electrical and biomechanical performance is the clear approach, while also researching new features such as biodegradability and regenerability can also reveal untapped opportunities.

## 5. Conclusions

In the ceaseless search for a minimal mismatch between the biologic and electronic, recent advances in hydrogel coatings and encapsulations have paved the way for hydrogel bioelectronics, which demonstrate a promising route to alleviate these adverse biomechanical interactions. Transcribing hydrogels with exceptional electrical and mechanical properties into a realistic working device is still a daunting challenge in the field, but also creates a stimulating new opportunity for seamless integration between biological and mechanical. Using GelMA hydrogel encapsulation, electronic devices can be successfully implanted and function in in vivo settings without triggering foreign body reactions.

Bioprinting GelMa is a process where the crucial temperature parameters must be accurately controlled and maintained at constant values because the material performs very differently in terms of mechanical properties with very small temperature variations. The material must be extruded at constant 26 °C then immediately cooled to 4 °C to maintain its desired shape until crosslinking and curing steps are completed.

The electrical conductivity of the encapsulated device with silicone is lower than that of the encapsulated device with GelMA, but this does not influence device function.

GelMA coating supports cell viability and cell integration within its structure, with lower immune response and better tissue integration.

Such augmented biomechanical and biochemical interfaces, along with superior biocompatibility, lower foreign body response and considerably extend the capabilities for bioelectronic applications [26]. Nonetheless, a successful implantation is ultimately determined by encapsulated device's technology in a certain field of interest. Such novel integration of hydrogel bioelectronics will require individual adjustments, including an increased demand of electrical or mechanical attributes, in addition to device assembly and manufacture.

**Author Contributions:** The authors have worked together for this study: biocompatibility tests: V.R. and R.G.; elaboration of the implantation procedure: I.L., A.M.O., V.C., T.P.N., S.L., C.S., R.I.T., D.C.D. and D.M.T.; resources, funding: C.A.M., B.F., E.F., M.D. and O.D.; in vivo tests: D.M.E., A.D. and A.M.; implantation surgery: I.L., A.M.O., V.C., T.P.N., R.C. and M.G.; writing—original draft preparation: D.C.D., V.C., C.A.M., E.F., M.D., O.I., G.I., O.D., S.R. and M.I.; writing—review and editing: C.N., V.C., I.R., C.M., E.F., M.D., O.I., G.I., O.D. and B.F.; supervision, D.C.D., C.N., V.C., I.R., C.M., E.F., M.D., O.I., G.I., O.D. and B.F. All authors have read and agreed to the published version of the manuscript.

**Funding:** This research was funded by the European Social Fund from the Sectorial Operational Programme Human Capital 2014–2020, through the Financial Agreement with the title "Training of PhD students and postdoctoral researchers in order to acquire applied research skills-SMART", Contract no. 13530/16.06.2022-SMIS code: 153734, and Project MicroNEx 20PFE/2021, financed by Program 1: Development of national research-development system, Subprogram 1.2. Institutional development projects – Projects for financing excellence in CDI (PDI-PFE), Ministry of Research, Innovation and Digitalization.

**Informed Consent Statement:** Not applicable.

**Data Availability Statement:** Not applicable.

**Acknowledgments:** Project ARMIN EEA Grant, EEA-RO-NO-2018-0390, nr. 8/2019. Contract 15/1.1.3H/342/390018/19.03.2021 (Moore4Medical-Accelerating Innovation in Microfabricated Medical Devices), SMIS code 137529.

**Conflicts of Interest:** The authors declare no conflict of interest.

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
