# Peer review of "Biocompatibility Analysis of GelMa Hydrogel and Silastic RTV 9161 Elastomer for Encapsulation of Electronic Devices for Subdermal Implantable Devices"

_coatings, doi:10.3390/coatings13010019_

Round 1
Reviewer 1 Report
The manuscript is quite interesting, but needs to be revised.
There is a strong imbalance between everything you say you have done and the results discussed.
In the results, there is no verification of the efficacy of GelMA synthesis or its encapsulation.
You say you have performed these processes, but there is a lack of experimental evidence to confirm this.
In fact, the results and discussion are poor because this part is missing.
In the results, it also seems strange to me to see 'cell viability' values with two decimal places.
The microscope photos are all before markers and it is not clear from the description which parts you are referring to, so you need to improve the description
Author Response
Thank you for the reviewers’ comments concerning our paper entitled “Biocompatibility analysis of GelMa Hydrogel and Silastic RTV 9161 Elastomer for encapsulation of electronic devices for sub-dermal implantable devices”. All comments are valuable and very helpful for revising and improving our paper, as well as the important guiding significance to our research. Now, we are submitting the revised version after seriously considering the comments of the Reviewers. We hope that this version will improve the quality of our manuscript and make it more acceptable for publication.
On behalf of my co-authors, I would like to clarify some of the points brought up by the Reviewers. We hope that the Reviewers and the Editor will be satisfied with our replies to the comments and our manuscript.

Reviewer 2 Report
Some specific comments are as follows:
Page 2, Line 65, “cultivated cell condition in vitro”…………………………Please mention the cell line name that will be used in the present study.
Page 2, Line 66, “in a living animal for 21 days and the extracted samples”…………………..What is the name of animal used? , samples are not extracted but you could refer to the selected tissue samples used in the histopathology technique.
Page 2, Lines 72-74 “by adding methacrylic anhydride in a 10% (w/v) suspension of gelatin in phosphate buffer at a rate of 0.5 mL/min under steady stirring.”…………….. This is not clear, adding methacrylic anhydride to what? To Gelatin in phosphate buffer 10%? Please clarify.
Also, please state the name of the manufacturer and city from where all used chemicals, equipment/instruments were sourced.
Please correct degrees in all the text to …………….℃
Page 2, Line 78, integration of LAP Photoinitiator………….this step is not clear, please clarify it more.
Page 2, Line 86, “The proposed elastomer RTV 9161 silicone rubber it’s a suitable material”…. The proposed elastomer RTV 9161 silicone rubber is a suitable material.
Page 2, Subtitle. Medical silicone. It is not clear… did you purchase them fully prepared, or authors prepared them by adding silicone rubber with catalyst?
Also, the properties of the medical silicone could be added in the introduction or within the discussion part if needed but not in the methodology part.
Page 3, Line 108, “needs to function is a sterilized medium”…… needs to function in a sterilized medium, sterilized area, or media? What medium?
Page 3, Line 110, In order to ensure that the medium is clean…….What do you mean by medium? The surrounding area? Please clarify.
Page 3, Line 110, “0.1 mm to 0.5 are fitted (fig. 1)”……… 0.1 to 0.5 mm are fitted (fig. 1).
Page 3, Line 111, After the material has been loaded into the syringe……..What material?
Page 3, Line 113, pneumatic pressure……Mention the pressure value and unit.
Page 4, in Figure 1. It will be good to add arrows on the figure showing labels that are mentioned within the text of bioprinter such as extruder, syringe, printing house, the printing bed and the printhead, ……etc.
Page 4, Line 144, is has been soaked in a ionic crosslinking solution…….it has been soaked in the crosslinking solution (mention this solution).
Page 5, Line 155, The materials were washed with sterile PBS………..specify the name of referred materials.
Page 5, Line 157, Mention the source of your cells. Also, mention that different groups were conducted in triplicate because author refer in Biocompatibility in vitro results n=3.
Page 5, Line 183, implanted in a sous scrofa subject to the experiment…….., implanted in a Sus scrofa.
What is the age, gender, need more information about used animal.
Page 5, Line 185, All procedures had been approved by the Faculty of Veterinary Medicine Ethics Committee………………What is the approval number and date? Refer here.
Page 5, Line 188, were surgically extracted………….. Please clarify this point….the devices were surgically extracted with the surrounding biological tissue for sectioning. The volume or approximate weight of the surrounding tissue used in fixation, embedding and sectioning . Before surgery, how did you kill the animal?
This part about histopathological technique needs reference.
Page 6, Line 210 & 211, Day 3: Control conditions, Day 6: Control conditions……..correct to …….control untreated cells.
Page 6, Figure 4, control group……more accurate to write……control (untreated cells).
Page 6, Figure 4, Gelma…………correct to……….. GelMA
Within the whole text, Control conditions……..correct to …….control untreated cells.
Page 7, Line 228, x20 and x40 zoom……………….it is not a zoom, correct to x20 and x40 magnification.
Figure 5. also correct the legend to magnification instead of zoom and add the scale value at the end of the legend. Also, the scale bar on the figure hard to be seen.
Page 7, Line 229, frequent macrophages, histiocytes and lymphocytes around fragments…….Add arrows, star, or lines on the related figure 5 to show those stated cells.
Page 8, Figure 6, correct the legend to magnification instead of zoom and add the scale value at the end of the legend. Also, the scale bar on the figure hard to be seen.
Page 8, Line 249, He stain…………………HE stain.
Page 9, Lines 255-279………………..Where are the references of those paragraphs?
The present results show high toxicity of the experimental materials in vivo and in vitro…..what is your discussion in this important point, is there mechanism for this toxicity, previous work in agreement to this work. In page 2, author state that “The proposed elastomer RTV 9161 silicone rubber it’s a suitable material that has medical grade appliances.” It is safe for use…..What is your comment on their toxicity? I didn’t find any explanation.
The author needs to discuss their results according to previous work, The discussion part is too poor in this, it should be rewritten in detail.
It is a good topic but Is it within the journal scope?
References are very few and not updated, the most recent references are 2017, the article needs more recent updated references.
- In the current state, there are more typographical errors. Few sentences need to be rewritten in more clear form with complete information, especially in the methodology. Therefore, the authors are advised to recheck the whole manuscript for improving the language and structure carefully.
Best
Author Response

(The authors gave the same response as above.)

Round 2
Reviewer 1 Report
no other comments
Author Response
Thank you for the reviewers’ comments concerning our paper entitled “Biocompatibility analysis of GelMa Hydrogel and Silastic RTV 9161 Elastomer for encapsulation of electronic devices for sub-dermal implantable devices”. All comments were valuable and very helpful for revising and improving our paper, as well as the important guiding significance to our research.
Reviewer 2 Report
Second review:
Most of the comments are done and corrected by the authors
But few specific comments are as follows:
Page 2, Line 71
and the selected tissue……………………….. and then the selected tissue samples around the implanted devices were used in the histopathological technique.
Page 3, Line 111
0.1 to 0.5 mm are fitted are fitted (fig. 1)………………………. 0.1 to 0.5 mm are fitted (fig. 1)
Page 9 Line 276
ad-verse effect…………………………..adverse effect
Page 9 Line 288
can deteriorates theits performance…………………………. can deteriorate their performance
Page 9 Line 291
and cell integra-tion………………………………… and cell integration
Page 9 Line 292
Noshida……………………Noshadi
Page 9 Line 292
[15]………………. [15].
Page 9 Line 297, 298, 300, 301, 303…………….correct
long-term im-plantation……………………… long-term implantation
the me-chanical and biological that simulates ……………correct…….the mechanical and biological that simulates ………………..
mechanical and biological???? what
GelMA encapsu-lation……………….. GelMA encapsulation
cell adhe-sion……………………………. cell adhesion
mechani-cal……………………… mechanical
Page 10 Line 342
mechanical proprieties………………… mechanical properties
Page 10 Line 343
26 ℃but…………………..26 ℃ but
Correct the newly added references according to the guideline of the journal
Finally, I suggest adding coatings within the text (not the conclusion only) to make your article related to the coatings journal and within scope.
Good work
best regards
Author Response

(The authors gave the same response as above.)
